# Oligodendroglial Epigenetics, from Lineage Specification to Activity-Dependent Myelination

**DOI:** 10.3390/life11010062

**Published:** 2021-01-15

**Authors:** Mathilde Pruvost, Sarah Moyon

**Affiliations:** Neuroscience Initiative Advanced Science Research Center, CUNY, 85 St Nicholas Terrace, New York, NY 10031, USA; mpruvost@gc.cuny.edu

**Keywords:** oligodendrocyte, oligodendrocyte progenitor cell, epigenetics, DNA methylation, histone, chromatin, long non-coding RNA, lamina

## Abstract

Oligodendroglial cells are the myelinating cells of the central nervous system. While myelination is crucial to axonal activity and conduction, oligodendrocyte progenitor cells and oligodendrocytes have also been shown to be essential for neuronal support and metabolism. Thus, a tight regulation of oligodendroglial cell specification, proliferation, and myelination is required for correct neuronal connectivity and function. Here, we review the role of epigenetic modifications in oligodendroglial lineage cells. First, we briefly describe the epigenetic modalities of gene regulation, which are known to have a role in oligodendroglial cells. We then address how epigenetic enzymes and/or marks have been associated with oligodendrocyte progenitor specification, survival and proliferation, differentiation, and finally, myelination. We finally mention how environmental cues, in particular, neuronal signals, are translated into epigenetic modifications, which can directly influence oligodendroglial biology.

## 1. Introduction

Oligodendrocytes (OLs) are glial cells of the central nervous system (CNS) that represent 20% of cells in an adult mouse brain and even up to 40% of neural cells in human neocortical regions [1,2]. While long considered simply as myelin-producing cells, many studies have since shown that they also interact with and respond to their environment, especially to neuronal signals [3]. Oligodendroglial cells in the forebrain originate from neural stem cells (NSCs) in three successive developmental waves, namely, ventral, dorsal, and eventually cortical, with the latest generating 50% of the total oligodendrocyte progenitor cells (OPCs) [4,5,6]. OPC specification in the spinal cord also follows two waves, from ventral to dorsal, resulting in approximately 80% of ventrally derived OPCs [7,8,9]. Once generated, OPCs can proliferate, migrate, and differentiate into mature myelinating OLs [10,11,12,13]. A proportion of these OPCs are also maintained as undifferentiated progenitors in the adult CNS, which directly interact with neurons and contribute to adult myelination and remyelination [14,15,16].

Oligodendroglial specification and then differentiation are regulated by the dynamic expression of transcription factors, which control the expression of lineage-specific genes (e.g., *Ascl1*, *Olig1*, *Sox10*) or myelinating genes (e.g., *Yy1*, *Myrf*) [17,18,19,20,21]. While these transcriptional events are tightly orchestrated, environmental cues are also critical in this process. For instance, OPCs have been shown to be highly responsive to neuronal activity, which can induce their proliferation or differentiation to mature oligodendroglial cells [22]. The response of OPCs toward external stimuli, and in particular, neuronal activity, is crucial for neuronal network organization and adaptivity. For example, myelin plasticity and remodeling in response to neuronal activity is known to be essential for learning and cognitive function of the CNS [23,24,25]. It is also crucial in response to injury and disease, when rapid OPC proliferation and differentiation form new myelin to support recovery and protect axons and neuronal function [26]. The integration of external cues, such as neuronal activity, into intrinsic signals is mediated by epigenetic modifications, which are known to control chromatin organization and, in turn, regulate gene expression. In particular, chromatin condensation and accessibility are regulated by DNA methylation, histone modifications, and chromatin remodelers, which interact with long non-coding RNA (lncRNA) and microRNA (miRNA), as well as nuclear organization via lamins [27]. Recently, the methylation of mRNA has also been described as an epigenetic modification, resulting in gene expression regulation at the translational level [28] (Figure 1).

In this review, we summarize how external signals communicate to the oligodendroglial cell lineage through epigenetic “messages.” In the first part, we focus on how each step of oligodendrocyte development, from lineage choice to myelination, is regulated by epigenetic modifications. We then focus on how environmental factors, and especially neuronal activity, can influence OPC gene expression by triggering epigenetic changes.

## 2. Epigenetic Modifications Involved in Chromatin Accessibility and the Regulation of Gene Expression

As previously mentioned, epigenetic modifications, in the current acceptation of the term, encompass the various mechanisms that modulate chromatin organization and gene expression. We discuss almost exclusively those modifications that have been shown to regulate cells within the oligodendroglial cell lineage (Figure 1).

### 2.1. DNA Methylation and Hydroxymethylation

DNA methylation consists of the addition of a methyl group, mostly occurring at the fifth carbon of cytosines in CpG dinucleotide sites in eukaryotic genomes [29]. In mammalian cells, the three major DNA methyltransferases (DNMTs) are DNMT1, DNMT3A, and DNMT3B [30,31]. Their catalytic activity is associated with the methylation of cytosines, which could prevent the access of transcription factors to their binding sequence and the associated recruitment of cofactors that modulate chromatin conformation. This modification results in transcriptional repression occurring in the gene promoter regions [32,33]. DNA hydroxymethylation, in contrast, is catalyzed by the ten-eleven translocation (TET) enzymes, mainly TET1, TET2, and TET3, and refers to the oxidation of 5-methylcytosine to 5-hydroxymethylcytosine [34,35]. DNA hydroxymethylation is mostly associated with transcriptional activation and is often detected at gene bodies and transcriptional starting sites [36,37].

### 2.2. Histone Post-Translational Modifications

A second epigenetic modification refers to the post-translational changes of amino acids in histones, which are the fundamental units of the nucleosome and are composed of an octamer of the four core histone proteins (H2A, H2B, H3, and H4). The tail of each histone is a site of post-translational modification, including methylation, acetylation, citrullination, SUMOylation, phosphorylation, ubiquitination, proline isomerization, and ADP-ribosylation [38]. The main epigenetic histone modifications described in oligodendrocytes occur on H3 or H4, at specific lysine (K) or arginine (R) sites, and consist of acetylation (ac), methylation (me1, me2, me3), or citrullination. The histone methylation of lysine residues is catalyzed by histone lysine methyltransferases (HMTs) (e.g., KMT2/MLL, SETDB1, and SV39H1/2, as well as SUZ12, EZH2, and EED, which are parts of the polycomb repressive complex 2, PRC2, in oligodendroglial cells) at lysine residues and by protein arginine methyltransferases (PRMTs) (e.g., PRMT1 and PRMT5 in oligodendroglial cells) at arginine residues [39,40,41,42,43,44]. Histone acetylation is catalyzed by histone acetyltransferases (HATs) (e.g., CREBBP and EP300 in oligodendroglial cells), and reversed by histone deacetylases (HDACs) (e.g., HDAC1, HDAC2, HDAC3, SIRT1, and SIRT2 in oligodendroglial cells) [45,46]. Histone modifications can either lead to gene activation (e.g., H3K9ac, H3K14ac, H3K27ac, H3K4me1, H3K4me3) or gene repression (e.g., H3K9me3, H3K27me3, H4R3me2s) [47,48,49,50,51,52,53]. Of special interest, H3K27ac and H3K4me1, as well as H3K4me3, have been strongly associated with gene activation at enhancers and transcription start sites, respectively, and are experimentally used to identify these regions [53,54]. Histone arginine citrullination, which is mainly catalyzed by peptidylarginine deiminase 2 (PADI2) in oligodendroglial cells, has a dual role in gene regulation. Citrullination is directly associated with gene activation, as it reduces proteins’ charges and histone-DNA histone interaction, favoring chromatin decondensation, while indirectly, it also blocks repressive methylation on arginine residues [55].

### 2.3. Chromatin Remodeling

Chromatin organization can further be rearranged by chromatin remodeling enzymes, which use ATP as an energy source to control chromatin accessibility [56]. Chromatin remodelers are classified into four different families, including the switch/sucrose-non-fermenting (SWI/SNF) complex (e.g., BRG1 in oligodendroglial cells), the inositol requiring 80 (INO80)/SWI-related (SWR) complex (e.g., EP400 in oligodendroglial cells), and the chromodomain-helicase DNA-binding (CHD) complex (e.g., CHD7 and CHD8 in oligodendroglial cells) [57,58,59,60,61,62]. Chromatin remodelers have the ability to regulate the activation or repression of gene expression by opening or closing the chromatin architecture [63].

Among the chromatin architectural proteins, the high-mobility group (HMG) superfamily comprises three families of proteins, namely, HMGA, HMGB, and HMGN, which dynamically bind to chromatin, promoting its decompaction and binding to nuclear regulatory factors [64,65].

### 2.4. 3D Chromatin Organization with Nuclear Lamina

Higher-order chromatin organization includes further genomic interaction with the nuclear lamina, which maintains nuclear structural integrity [66]. Lamina-associated domains are usually enriched for the repressive histones (e.g., H3K27me3 and H3K9me3) characteristic of heterochromatin [67]. Two protein families have been shown to form the nuclear lamina: A-type lamins (lamin A and C) and B-type lamins (lamin B1 and B2) (e.g., LMNB1 in particular in oligodendroglial cells) [66,68].

### 2.5. Post-Transcriptional Modifications

Long non-coding RNA (lncRNA) and microRNA (miRNA) are transcribed but non-translated RNAs that regulate gene expression at transcriptional and post-transcriptional levels, in particular, during CNS development [69,70,71,72]. The lncRNA are more than 200 nucleotides long, while the miRNA, which are processed by RNA polymerase II and DICER, are typically 20–25 nucleotides long. In addition to the ability to bind and regulate chromatin accessibility, lncRNA and miRNA can affect and regulate the expression of nascent mRNA transcripts that are encoded at distant genome loci due to post-transcriptional mechanisms [73].

Recently, additional chemical modifications have been identified at the messenger RNA (mRNA) level [74,75]. The most common in mammalian cells, namely, methylation of N^6^-methyladenosine (m^6^A) mRNA, is involved in post-transcriptional regulation of gene expression by influencing mRNA stability, translation, localization, and splicing [76,77,78]. In oligodendroglial cells, METTL14 has been described as the core component of the m^6^A methyltransferase complex, while FTO is the main mRNA demethylase [79,80].

An accumulation of evidence has revealed that all layers of epigenetic modifications described in this section are involved in the oligodendroglial cell lineage, from specification choice to survival and differentiation.

## 3. Epigenetic Marks with Roles in Oligodendroglial Cell Lineage

NSCs in the developing and adult mammalian brain harbor the ability to self-renew and to generate neurons, astrocytes, and oligodendrocytes [81]. The differentiation of NSCs occurs in response to extracellular signals, along with the interplay between dynamic epigenetic modifications and lineage gene expression regulation [82,83]. Lineage specification requires both the activation of lineage genes and the repression of alternative lineage genes. In this section, we will focus on how the transition from NSCs to OLs is regulated by successive waves of DNA methylation and hydroxymethylation, as well as by histone modifications, chromatin remodelers, microRNAs, and lncRNA (Figure 2 and Table 1).

### 3.1. DNA Methylation Waves in NSC Differentiation to OPCs

The specification of NSCs from embryonic stem cells (ESCs) toward neurogenesis and gliogenesis occurs through specific methylation patterns. A study from Sanosaka et al. suggests that the demethylation of neuron-specific genes first occurs as ESCs transition to NSCs [84], while methylation on glial promoters, such as glial fibrillary acidic protein (GFAP), is maintained to suppress its expression [85]. Gliogenic transition is then favored and occurs via the demethylation of glial gene promoters and genes involved in gliogenic pathways, as well as the de novo methylation of neuronal genes [84,85,86,87,88,89,90]. Subsequently, the de novo methylation of astrocytic genes occurs during the specification of oligodendrocytes [86,91], along with the hydroxymethylation of oligodendrocyte genes, such as *Olig1*, *Sox10*, and *Id2/4* [92,93].

**Table 1 life-11-00062-t001:** List of epigenetic marks and their roles during OPC specification.

DNA Modification	Enzyme/Mark	Role	Targeted Genes or Functions	Model	Methods	References
DNA methylation		Negative role in NSCs differentiation to glia	Repression of astrogliogenesis (*Stat3* binding element in *Gfap* promoter)	In vitro neuroepithelial cells	BS PCR on neuroepithelial cells	Takizawa et al. [85]
	DNMT1	Negative role in NPCs differentiation to glia	Control of the timing of astrogliogenesis by repression of astrocyte-specific genes (*Gfap*, *Stat1*)	Nestin-cre;Dnmt1flox NPCs	BS PCR on Nestin-cre;Dnmt1flox NPCs	Fan et al. [86]
		Positive role in NSCs differentiation to glia	Repression of neuron-specific genes (*Dlx1, Dlx2, Trb1*)	Sox2-EGFP mice	WGBS on NS/PCs from Sox2-EGFP transgenic mice	Sanosaka et al. [84]
	DNMT1	Positive role in NSCs specification to OPCs	Repression of neuron-specific (*Ndrg4*, *Camk1*, *Ephb2*) and astrocyte-specific (*Aldh1l1*, *Pax6*, *Rfx4*) genes	Olig1cre;Dnmt1flox mice	RNA-Seq and ERRBS on sorted neonatal OPCs and OLs, RNA-Seq on sorted Olig1cre;Dnmt1flox OPCs	Moyon et al. [91]
DNA demethylation		Positive role in ESCs transition to NSCs	Activation of transcription factors (*Sox2*, *Sox21*, *Ascl1*)	Sox2-EGFP mice	WGBS on NS/PCs from Sox2-EGFP transgenic mice	Sanosaka et al. [84]
		Positive role in NSCs differentiation to glia	Activation of gliogenic promoters (*NFI*, *Tcf3*, *Gfap*, *Kcnj10*, *Sox8*)			
			Activation of gliogenic promoters (*Stat3* binding site in *Gfap* promoter, *Aldoc*)	In vitro NPCs and astrocytes from mice	MIAMI and BS on NPCs and astrocytes	Hatada et al. [87]
DNA hydroxymethylation	TET1	Positive role in NSCs differentiation to OPCs	Activation of OL genes (*Olig1*, *Sox10*, *Id2/4*)	Olig1cre;Tet1flox mice	hMeDIP-Seq on in vitro neonatal NSCs and OPCs, RNA-Seq on in vitro Olig1cre;Tet1flox OPCs	Zhang et al. [92]
** Histone Modification **	** Enzyme/Mark **	** Role **	** Targeted Genes or Functions **	** Model **	** Methods **	** References **
Histone methylation	PRC2 (EZH2) (H3K27me3)	Positive role in NSCs differentiation to OPCs		In vitro primary cultures + Olineu + *Ezh2* expression vector/shRNA		Sher et al. [94]
			Repression of neuronal (*NeuroD2*, *Tlx3*), astrocytic (*Tal1*), OL (*Olig2*, *Pdgfra*, *Nkx2.2*) genes	In vitro NSCs and OPCs + shRNA	ChIP-Seq on in vitro NSCs and OLs	Sher et al. [95]
	PRC2 (EED) (H3K27me3)	Positive role in astrocyte–OPC fate switch		Olig1cre;Eedflox and PdgfracreRT;Eedflox mice		Wang et al. [96]
	H3K27me3	Positive role in NSCs differentiation to OPCs	Repression of global lineage alternative choice	In vitro OPCs; Cnp-EGFP mice	ChIP-Seq on in vitro OPCs and OLs (H3K27me3)	Liu et al. [97]
	PRMT1	Positive role in NSCs differentiation to OPCs		Nestin-cre;Prmt1flox mice		Hashimoto et al. [98]
Histone (de)acetylation	CBP (H3K9/K14ac)	Positive role in NSCs differentiation	Sequential activation of promoters of neuronal (*Tuba1a*), astrocytes (*Gfap*), OL (*Mbp*)	In vitro cortical precursors + siRNA/inhibitors; cbp+/− mice	ChIP-qPCR on cbp+/− cortices	Wang et al. [99]
	HDACs	Positive role in oligodendrogenesis		In vivo HDAC inhibition in rats		Liu et al. [100]
	HDAC1/HDAC2	Positive role in Shh-induced oligodendrogenesis	Repression of genes associated with Notch signaling (*Hey1*, *Hey2*) and Wnt signaling (*Tbx3*)	In vitro OPCs, Olineu cells + shRNA/inhibitors	ChIP on in vitro Olineu and GeneChip on in vitro OPCs	Wu et al. [101]
	HDAC2 (H3K9deac)	Negative role in NSCs differentiation to OPCs	Repression of oligodendroglial differentiation genes (*Sox10*) in the presence of thyroid hormone	In vitro NSCs, Olineu, OPCs + siRNA/inhibitors	ChIP-Seq on in vitro NSCs	Castelo-Branco et al. [102]
	HDAC3	Positive role in astrocyte–OL fate switch	Activation of enhancers of OPC genes (*Olig2*, *Ng2*) and repression of astrogliogeneis genes (*Stat3*) and neuronal genes (*Bdnf*)	In vitro astrocytes and OPCs + inhibitors/expression vectors; Olig1Cre;Hdac3flox, PDGFRaCreERT2;Hdac3flox; Syn1Cre;Hdac3flox mice	ChIP-Seq on in vitro OPCs and OL; RNA-Seq on optic nerves from Olig1cre;Hdac3flox mice	Zhang et al. [103]
	SIRT1 (H3K9deac)	Negative role in NSCs differentiation to OPCs	Repression of differentiation to OPCs (deacetylation at the *Pdgfrα* promoter)	In vitro OPCs, NS/PCs + inhibitors; NestinCre;Sirt1flox mice	ChIP-qPCR on in vitro NS/PCs	Rafalski et al. [104]
** Chromatin Organization **	** Enzyme/Mark **	** Role **	** Targeted Genes or Functions **	** Model **	** Methods **	** References **
Chromatin remodelers	HMGA1, 2	Negative role in NPCs differentiation to glia	Repression of astrogenic transition	In vitro and in vivo NPCs + shRNA/overexpression		Kishi et al. [105]
	HMGB1, 2, 3, 4		Dynamically expressed in NSCs	In vitro NSCs, Nestin-GFP mice, HMGB2−/− mice	IHC, qPCR, shotgun proteomics on in vitro NSCs	Abraham et al. [106]
	HMGB2	Possible role in NSCs proliferation and maintenance		In vitro NSCs, Nestin-GFP mice, HMGB2−/− mice	IHC, qPCR, shotgun proteomics on in vitro NSCs	Abraham et al. [106]
		Role in neuron–glia fate switch		In vitro NS/PCs, HMGB2−/− mice		Bronstein et al. [107]
	HMGB4	Negative role in NSCs differentiation to OPCs	Regulation of neuronal, astrocyte, oligodendrocyte genes (*Fabp7*, *NeuroD1*, *Gfap*, *Ppp1r14a*)	In vitro neurons, neurospheres and various cell lines + lentivirus; HMGB4 Vivo-Morpholinos	Microarray on HMGB4-EGFP over-expressing HEK 293T cells	Rouhiainen et al. [108]
	HMGN family	Positive role in neuron–glia fate switch	Modulation of the response to gliogenic signals	In vitro NPCs + shRNA/overexpression	qPCR, IHC, BS and microarray on in vitro NPCs	Nagao et al. [109]
	BRG1	Positive role in NSCs maintenance and gliogenesis	Repression of neuronal differentiation in NSCs	In vitro NSCs, NestinCre;Brg1flox mice	IHC; DNA microarray on CNS tissues of NestinCre;Brg1flox mice	Matsumoto et al. [110]
		Positive role in NSCs differentiation to neurons	Activation of neuronal genes (*Ngn* and *NeuroD*)	In vitro pluriopotent P19 cells + plasmids, Xenopus + Brg1 morpholino	IHC	Seo at al. [111]
		Limited role in NSCs		zebrafish + Brg1 morpholino	IHC, ISH	Gregg et al. [112]
** Post-Transcriptional Modification **	** Enzyme/Mark **	** Role **	** Targeted Genes or Functions **	** Model **	** Methods **	** References **
Long non-coding RNA	*lnc-158*	Positive role in NSCs differentiation to OPCs	Activation of OL genes (*Cnp*, *Mbp*, *Mag*, *Osp*)	In vitro NSCs + *lnc-158* overexpression and siRNA		Li et al. [113]
	*lnc-OPC*	Positive role in NSCs differentiation to OPCs	Activation of OL genes (*Mbp*, *Plp1*, *Cnp*)	NSCs + shRNA	ChIP-Seq and RNA-Seq on in vitro NSCs	Dong et al. [114]
	*Neat1*	Positive role in NSCs differentiation to OPCs	Activation of OL genes (*Olig1*, *Olig2*, *Gpr17*, *Sox8*)	Neat1−/− mice	ChIP-Seq on human tissues; RNA-Seq on Neat1−/− mouse brains	Katsel et al. [115]
	*Sox8OT*	Possible role in NSCs differentiation to OPCs	Via *Sox8* activation	Descriptive study		Mercer et al. [116]
MicroRNA	*miR-124*	Positive role in NSCs differentiation to neurons	Repression of *Ezh2* expression	In vitro NSCs + N2a neuroblastoma + P19 cells + overexpression	Microarray on N2a cells	Neo et al. [117]
	*miR-153*	Negative role in NSCs differentiation to glia	Repression of gliogenic genes (*Nfia/b*)	In vitro ESCs + shRNA + artifical miRNAs		Tsuyama et al. [118]
	*miR-17/106*	Positive role in NSCs differentiation to glia	Activation of gliogenesis (*p38*)	in vitro ESCs and mouse embryos + lentivirus		Naka-Kaneda et al. [119]

BS: bisulfite sequencing, ChIP: chromatin immunoprecipitation, ERRBS: enhanced reduced representation bisulfite sequencing, hMeDIP-Seq: hMeDIP: hydroxyMethylated DNA immunoprecipitation; IHC: immunohistochemistry; ISH: in situ hybridization; NPCs: neural precursor or progenitor cells, NS/PCs: neural stem/progenitor cells; MIAMI: microarray-based integrated analysis of methylation by isoschizomers, qPCR: quantitative polymerase chain reaction; Seq: sequencing, WGBS: whole-genome bisulfite sequencing.

### 3.2. Histone Modifications

#### 3.2.1. Repression of Lineage-Specific Genes in NSCs by Histone Methylation

Lineage determination is also regulated by histone modifications, such as H3K27me3, which is catalyzed by the polycomb repressive complex 2 (PRC2) and is constituted of the enhancer of the zeste homolog 2 (EZH2) and embryonic ectoderm development (EED). EZH2 and EED are highly expressed in proliferating NSCs and target genes related to developmental processes and neurogenesis [95,96]. As NSCs differentiate into OPCs, EZH2 and EED expression remain high, while their levels decrease in cells transitioning to neuronal and astrocyte lineages [94,96]. Moreover, the ChIP-sequencing of H3K27me3 in postnatal-day-1 rat OPCs highlights its genomic regulation of genes involved in the “global alternative lineage choice” [97]. Indeed, conditional ablation of *Eed* favors the differentiation into astroglial cells, to the detriment of the oligodendroglial lineage [96]. OPC lineage progression also depends on arginine methylation, as the conditional knockdown of *Prmt1* in NSCs drastically reduces the number of OLs in mice [98].

#### 3.2.2. Histone Acetylation and Deacetylation Regulate the Lineage Specification

The normal differentiation of cortical precursors in vitro and in vivo in mice is accompanied by sequential histone acetylation and the subsequent activation of promoters of neuronal and astrocyte genes, then postnatally in oligodendrocyte genes, such as *Mbp* and *Plp*. The key histone acetyltransferase in this process is the CREB binding protein (CBP), which regulates H3K9/K14 acetylation [99]. Indeed, blocking histone deacetylation in rats reduces oligodendrogenesis and favors the differentiation of cells along alternative lineage choices [100]. However, histone deacetylation has also been shown to be involved in the regulation of neural progenitors [120]. In particular, HDAC2 and HDAC3 associate with key genes regulating the differentiation of NSCs [102,121]. In NSCs, HDAC3 acts first as a repressor of neuronal differentiation in cortical NSCs [102], then antagonizes astrogliogenesis by inhibiting the acetylation levels of *Stat3*, which is a core effector in the Janus kinase (JAK)–STAT pathway [103]. Astrocytic genes are also repressed by HDAC1 and HDAC2 in vitro during Sonic Hedgehog (Shh)-induced oligodendrogenesis [101]. Finally, the cooperation of HDAC3 with the histone acetyltransferase p300 targets and activates enhancers of the OPC genes, such as *Olig2* [103]. Inversely, the differentiation to OPC is prevented by HDAC2 and sirtuin 1 (SIRT1), which, at least in part, can inhibit specific OL differentiation genes, such as the key transcription factor *Sox10* [102,104].

### 3.3. Chromatin Reorganization during the Differentiation from NSCs to OPCs

Chromatin structure and organization in NSCs changes during their specification throughout brain development to generate the major cell types of the CNS. These dynamic chromatin states are modulated by chromatin architectural proteins, such as the HMG proteins, and chromatin modifiers, such as BRG1. In the early stages of NSCs characterized by a neurogenic potential, HMGA1 and 2 proteins are highly expressed and mediate the global chromatin opening. As the levels of HMGA proteins decrease, chromatin becomes more condensed in a stage-dependent manner that allows for an astrogenic transition [105]. All four mammalian forms of HMGB (HMGB1, 2, 3, and 4) are expressed in proliferating NSCs [106] with specific roles described for HMGB2 and HMGB4 in the neurogenic-to-gliogenic fate transition [107,108]. Members of the HMGN family (HMGN1, 2 and 3) also positively regulate the neuron–glia fate switch [109].

BRG1, within the SWI/SNF-related chromatin remodeling complex, is a critical regulator of NSC specification by repressing neuronal differentiation, while favoring gliogenesis and differentiation in mammalian neural development. In mice, the ablation of *Brg1* specifically in NSCs does not impact the initial neuronal differentiation but abolishes glial generation and differentiation, as seen by a dramatic decrease in astrocyte, oligodendrocyte progenitor, and myelin protein markers in late embryonic stages [110]. However, the function of BRG1 appears different in lower vertebrates, as the inhibition of BRG1 blocks neuronal differentiation in *Xenopus* [111] and has restricted effects in retinal ganglion cells in zebrafish [112].

### 3.4. Post-Transcriptional Modifications

Several miRNAs play key roles in neural lineage development. For instance, *miR-124* is necessary for fate specification into neurons by reducing the expression of EZH2 [117]. Other miRNAs are necessary for the neurogenic-to-astrogenic transition, such as *miR-153*, which regulates the acquisition of gliogenic competence [118], or *miR-17/106* [119]. To our knowledge, no miRNA has been described as being specifically required for the differentiation from an NSC to an OL.

Dong and collaborators have investigated whether lncRNAs are involved in the regulation of NSC differentiation into OPCs [114]. They identified *lnc-OPC*, a specific and highly expressed lncRNA in OPCs, which is critical for cell fate determination. In vitro loss- and gain-of-functions experiments that targeted *lnc-OPC*, as well as *Sox8OT*, *Neat1* and *lnc-158*, have highlighted the positive prominent role of lncRNAs for oligodendroglial specification [113,114,115,116].

## 4. Epigenetic Marks Maintain an Oligodendroglial Progenitor Cell Pool

In the CNS, oligodendroglial cells are highly sensitive to their environment, which influences their survival and proliferation rate as immature OPCs and/or their differentiation capacities into myelinating OLs. Overall, this tight regulation maintains a relative homeostatic pool of OPCs throughout time and space in both the developmental and adult CNS [122]. In particular, OPCs proliferate in response to chemical cues, such as mitogens, growth factors, and cytokines [10,123,124,125,126,127], and to neuronal cues, such as electrical activity [22,128,129,130,131]. OPCs are now thought to translate these specific extracellular cues into intrinsic signals that affect survival and proliferation via the regulation of epigenetic modifications (Figure 2 and Table 2).

### 4.1. Apoptosis vs. Survival

Increasingly, evidence has revealed the heterogeneity of the OPC properties between the developmental and adult OPC pools, but also according to their CNS localization. OPC survival has been shown to be variably associated with DNA and histone modifications and chromatin remodeling, depending on time and space.

During development, the ablation of *Dnmt1* or *Dnmt3a* in neonatal OPCs tends to induce DNA damage and decreases OPC survival in vitro [91,132]. However, these effects were not observed after the ablation of *Dnmt1* and/or *Dnmt3a* in adult OPCs during remyelination experiments [142]. OPC survival during development has also been associated with the PRMT5-dependent H4R3me2s mark, as the ablation of PRMT5 in OPCs could activate p53 pathways and increase apoptosis [134]. In addition, neonatal non-proliferative OPCs are protected from apoptosis by the chromatin remodelers EP400 and CHD7, the latest known remodelers to control chromatin closing and *p53* transcriptional repression [61,139]. In adult OPCs, cell survival is partially regulated by the CHD8, which shares many common binding sites with CHD7 [61]. Indeed, the global or oligodendroglial-specific ablation of *Chd8* in OPCs results in increased apoptosis, in particular, in adult spinal cord tissues, but not during development or in the brain [138].

### 4.2. Cell Cycle and Proliferation

The nuclei of proliferative OPCs are mostly euchromatic and characterized by a relaxed and transcriptionally competent chromatin structure that is enriched for permissive marks, such as histone acetylation (e.g., H3K9ac and H3K14ac) [104,135,143]. The OPC cell cycle is also regulated by the oncogene transcription factor cMyc in response to platelet-derived growth factor (PDGF) and fibroblast growth factor (FGF) mitogens [135,144]. cMyc recruits histone acetyltransferases and the ablation of *cMyc* in vitro can directly decrease H3K9ac and H3K14ac in OPCs [135,145,146,147]. This suggests that, on top of acting as a DNA binding transcription factor, cMyc is capable of translating mitogenic extracellular stimuli into epigenetic signals in proliferative OPCs.

Evidence suggests that the epigenetic regulation of OPC proliferation might differ depending on the CNS region. For example, differences in chromatin remodeling have been noted in brain and spinal cord tissues. The ablation of *Chd7* or *Chd8* in OPCs does not lead to an increased number of OPCs, since, despite an increase in cell cycle and proliferative genes, this also leads to an increase in apoptosis, at least in brain tissues [59,61,138]. Surprisingly, following *Chd7* or *Chd8* ablation, OPC proliferation is not increased but reduced in spinal cord tissues, where apoptosis is not perturbed [137,138].

OPC proliferation and cell cycle exits appear to be regulated by several epigenetic marks, including repressive ones, such as DNA and histone methylation. For example, the ablation of *Dnmt1* in neonatal OPCs revealed reduced DNA methylation and defective gene repression, in particular, at cell cycle genes [91]. This result was not replicated in vivo in adult OPCs, suggesting once again that there is an age-dependent epigenetic regulation of transcriptome in OPCs [142]. However, the presence of DNA modification alone is not sufficient to induce precocious proliferation. Ablating DNMT1- or DNMT3A-mediated DNA methylation or EED-mediated histone methylation in OPCs does not result in ectopic proliferation in vitro or in vivo, and even slightly negatively perturbates OPC proliferation during development [91,96,142]. Inversely, TET1-mediated DNA hydroxymethylation does not seem to affect neonatal or adult OPC proliferation in vitro or in vivo [92,133].

## 5. Epigenetic Marks Regulate the Oligodendroglial Progenitor Cell Differentiation into Mature Oligodendrocytes

One of the main and most studied functions of neonatal and adult OPCs is their ability to differentiate into mature OLs. OPC differentiation is in part an intrinsic propensity but is also regulated by environmental and neuronal cues that can influence epigenetic modifications (Figure 2 and Table 3). Recent studies highlighted the inter-neuronal (e.g., myelinated or non-myelinated axons) and intra-neuronal (e.g., variable size of internodes) heterogeneity of myelination in the CNS, which appeared to be essential for global brain connectivity and function, and which could reflect the heterogeneity of the OPC population and/or their environment [148,149,150,151,152,153].

### 5.1. Global DNA Demethylation Is Associated with OPC Differentiation

Early evidence showed that OPC differentiation is associated with demethylation of a specific myelin gene, *Mag*, during rat development [169]. A more global demethylation during OPC differentiation that is associated with permissive gene expression was further confirmed by DNA methylation whole-genome sequencing. During development, OPC differentiation into OL is correlated with the decreased DNA methylation and increased DNA hydroxymethylation of genes involved in lipid synthesis and myelin formation [91,92]. However, because of its role in OPC proliferation and survival, the ablation of DNA methyltransferases in neonatal OPCs does not induce early differentiation, but in contrast, results in global hypomyelination of the CNS [91,132]. Indeed, the ablation of DNA methyltransferases in developmental and adult post-mitotic OPCs has a slight effect on myelination and remyelination [91,142]. Recent studies also highlight an age-dependent role of demethylation on OPC differentiation, suggesting a more important role of DNA hydroxymethylation in adult oligodendroglial cells [92,133]. In particular, TET1-mediated DNA hydroxymethylation targets genes involved in the late stages of OPC differentiation, such as biosynthesis and neuroglial communication [92,93,133]. The ablation of *Tet1* in adult OPCs is sufficient to reduce DNA hydroxymethylation at these specific genomic sites, downregulating their expression and blocking OL late differentiation [133]. In old mice, lower TET1 expression and decreased DNA hydroxymethylation levels could be directly associated with the delayed remyelination observed in aging [133]. While the downregulation of TET enzymes in vitro has been shown to affect neonatal OPC differentiation, this effect is still being examined in developmental studies in vivo [78,92,133]. Overall, these studies suggest a dual role for DNA modifications in oligodendroglial cells, balancing both the methylation and demethylation of specific genomic regions at different ages and different stages of the differentiation process.

### 5.2. Chromatin Modifications Are Involved in OPC Differentiation

#### 5.2.1. Chromatin Reorganization Allows for the Accessibility of Differentiation Gene in OPCs

At the chromatin level, OPC differentiation is mainly associated with open conformation at discrete loci, which is reflected by chromatin remodeling, histone modifications, and nuclear lamin reorganization, allowing for access of transcription factors to specific genes that are characteristic of the differentiated state. For example, the chromatin remodelers BRG1 and EP400 have been shown to be essential for OPC differentiation and myelination, at least during development when targeting early lineage [57,62,139]. However, they seem to be dispensable for later differentiation stages and during myelin maintenance [57,139]. Both BRG1 and EP400, in association with OLIG2, directly bind to differentiation genes, such as *Myrf* and *Sox10*, especially at enhancers and transcription start sites, which are characterized by the permissive histone marks H3K27ac and H3K4me3, respectively [57,59,62,138,139]. They also share similar chromatin occupancies with CHD7 and CHD8, which are essential for OPC differentiation, during developmental and adult (re)myelination [59,61,138]. In particular, the ablation of *Chd8* in neonatal OPCs results in massive hypomyelination of the CNS, leading to seizures and eventually death of the mice at postnatal day 21 [138]. Interestingly, CHD8 can itself directly recruit KMT2/MLL, a histone lysine methyltransferase that is responsible for the addition of H3K4me3 and the subsequent activation of oligodendroglial genes (i.e., *Olig1/Olig2*, *Sox10*, *Myrf*) [138].

#### 5.2.2. Repressive Marks Regulate the Transition from OPC Proliferation to Differentiation

OPC differentiation is also dependent on marks that are generally associated with gene repression, such as histone methylation, deacetylation, and citrullination. Indeed, the ablation or inhibition of *Prmt5*, *Ezh2*, *Eed*, *Hdac1/2*, or *Padi2* in OPCs results in defective differentiation and myelination [55,96,97,121,132,134,136,143,155,156,157,159]. These marks mainly regulate the downregulation of OPC-specific inhibitors of differentiation (such as *Id2/Id4*) or cell cycle (such as *Cdk4/6*, *Cxcl2/5/10/14*) genes, and therefore, are often essential for OPC cell cycle exits and early OL differentiation, but less involved in myelin maintenance [55,96,97,155]. These enzymes, such as HDACs, PRMT5, and PADI2, can also modify non-histone targets, altering the function or localization of oligodendroglial proteins or transcription factors (i.e., OLIG1, alpha-tubulin, PDGFRα, or myelin proteins) [55,154,158,160,170]. Because of steric constraints, histone modifications can also be dependent on or exclusive of each other. For example, the ablation of *Prmt5* in OPC results in decreased symmetric H4R3me2s, allowing for H4K5 acetylation and preventing differentiation and myelination. The addition of histone acetylation inhibitors in vitro is sufficient to rescue OPC differentiation, even without PRMT5 [134].

Eventually, repressive marks are also associated with the nuclear lamina, which maintains mainly repressive heterochromatin at the nuclear periphery. Recently, LMNB1 has been associated with oligodendroglial maturation genes (i.e., myelin genes and cholesterol synthesis pathways), which appeared to be sufficient to block OPC differentiation in vitro [68,171].

### 5.3. Post-Transcriptional Modifications That Are Essential for OPC Differentiation

In addition to chromatin condensation and conformation, the OPC differentiation program also depends on post-transcriptional modifications. Recent models using the ablation of *Dicer* (required for the generation of functional microRNA) or the de novo analysis of oligodendroglial transcriptomic datasets identified microRNAs and lncRNAs that are essential for OPC differentiation. For example, *miR-219*, *miR-338*, *miR-23*, and *miR-32*, as well as *lnc-OL1*, *lnc-158*, and *Neat1*, promote differentiation, while *miR-27a*, *miR-212*, and *miR-125-3p* inhibit differentiation [113,115,140,161,162,163,164,165,166,167,171,172,173,174]. Interestingly, these post-transcriptional signals directly regulate some chromatin-modifying genes, suggesting feedback from miRNA and lncRNA regarding chromatin conformation. For example, *miR-23* has been shown to suppress LMNB1 expression, thus rescuing OPC differentiation in vitro [171]. Similarly, *lnc-OL1* can directly interact with SUZ12, a part of the histone methylation complex PRC2, with both being required for OPC differentiation during development and repair [161,175].

Recently, studies have identified the essential role of RNA methylation on OPC differentiation. The ablation of the methyltransferase *Mettl14* (“writer”), the demethylase *Fto* (“eraser”), or the m^6^A “reader” *Prrc2a* in the oligodendroglial cell lineage all resulted in a lower number of mature OLs and global hypomyelination of the CNS [141,168]. In addition to the downregulation of several myelin genes (i.e., *Mbp*, *Mog*, *Mag*), many histone post-translational modification readers and writers (i.e., HMTs, HDACs) are also dysregulated when METTL14 is lacking, suggesting here again the potential feedback from m^6^A-modified mRNA to histone modifications [168].

## 6. Epigenetic Marks Modulate Myelination and Myelin Remodeling

Finally, in addition to timely OPC differentiation into mature OLs, recent studies have highlighted the importance of correct myelination for neuronal connectivity. This includes myelin ensheathment, compaction around the axon, and internode sizes. This is not limited to developmental myelination as myelin is constantly remodeled in adult CNS, especially during learning and when facing depressive or social isolation experiences [24,25,176,177,178,179,180,181] (Table 4).

A few epigenetic marks have been associated with late myelination processes. Compared to other chromatin remodelers, CHD7 binds preferentially to myelinogenesis (i.e., *Mbp*, *Plp1*, *Cnp*) and lipid metabolism genes (i.e., *Enpp2*, *Nfya*, *Elovl7*), which would suggest involvement in late OPC differentiation and myelination [59]. The genetic ablation of *Tet1* in OPC induces defective remyelination, which is characterized by swellings in adult CNS after injury. TET1-mediated hydroxymethylation tends to target genes related to late myelination, as well as neuroglia communication genes involved in ion exchange and the maintenance of a tight space between the axon body and the myelin membrane (i.e., *Slc12a2*) [92,133]. METTL14 and m^6^A RNA affects the alternative splicing of some paranodal genes, such as glial neurofascin 155 [168]. In *Mettl14* mutants, Xu et al. noticed increased nodal/paranodal spaces, which were associated with lower numbers of nodes, suggesting a role for mRNA methylation on internode length [168].

Because the myelination and myelin remodeling processes have not been analyzed in detail yet, additional epigenetic modifications are to be identified in this later process to finely tune the ensheathment, internodal length, and myelin compaction.

## 7. Epigenetic Changes Translate Environmental Cues into Intrinsic Signals in Oligodendroglial Cells

OPC differentiation is an intrinsic program that is capable of myelinating dead axons or nanofibers in vitro [182,183,184]. Meanwhile, OPCs and OLs are also highly sensitive to their environment, which can modulate their physiology, in part by epigenetic mechanisms. Perhaps most interestingly, OPC proliferation and differentiation have been shown to be partially regulated by neuronal signals, including neurotransmitters and axonal activity, some of which could translate into epigenetic changes in oligodendroglial cells.

### 7.1. Chemical and Physical Cues

The OPC-to-OL switch is modulated by pro-proliferation or pro-differentiation chemical cues. For example, growth factors (e.g., PDGF and FGF) can induce cMyc expression in OPC, which can directly recruit histone acetyltransferases, upregulating genes involved in OPC proliferation [135,144,145,146,147]. Inversely, the addition of T3, known to trigger OPC differentiation in vitro, has been shown to induce *Brg1* expression, which would favor the accessibility of differentiating and myelinating genes [62]. Physical constraints in vitro and tissue stiffness in vivo have also been shown to influence OPC proliferation and differentiation [184,185,186]. Hernandez and colleagues showed that, in vitro, these mechanical stimuli are transduced via cytoskeleton and nucleoskeleton complexes into nuclear cues, resulting in histone methylation changes [185].

### 7.2. Neuronal Activity

#### 7.2.1. Oligodendrocytes Respond to Neuronal Activity

Neuronal activity plays a central role in brain function by shaping developmental circuits and providing effective transmission of the information necessary for behavioral and cognitive functions. OLs are key contributors to this process as they ensheath axons with their processes, forming the multilamellar membrane, myelin, which allows for energy-efficient saltatory conduction [187]. Several lines of evidence now demonstrate a reciprocal relationship between neurons and OLs, with OLs providing metabolic support to axons, which is crucial to their survival [3,188,189], and neurons being able to regulate OPC proliferation and differentiation [14,22,23,128,131,190,191,192], as well as myelin formation, repair, and remodeling [149,193,194,195]. A variety of in vitro and in vivo approaches, combined with experimental modulation of neuronal activity, have been used to characterize this relationship, such as the use of myelinating co-cultures [129,196,197,198], electrophysiology [199], enriched or deprived environments [149,177,178,200,201], motor and learning tasks [23,25,202,203], electrical stimulation [131], pharmacogenetic and chemogenetic tools [191,204], and optogenetics [22,194,205]. While neuronal activity can undeniably control OL biology, myelination can yet occur in the absence of such signals [206,207]. Interestingly, the pattern of neuronal activity is also thought to differentially impact OL and myelination [14,208].

Over the years, various signaling mechanisms have been proposed as the means for OLs to respond to neurons, with some requiring direct contact and others thought to occur via intermediate messengers to OL membrane receptors and ion channels. The identified pathways involve contact-mediated factors, such as N-cadherin [209,210]; trophic and growth factors, such as Neuroligin-3 and BDNF [211]; calcium signaling [212]; vesicular and non-vesicular release of neurotransmitters [196,213,214,215,216].

Single-cell studies have begun to identify transcriptional changes induced in the CNS in response to neuronal activity, including in OLs. In the mouse visual cortex, dynamic transcriptional changes, including in OL gene networks, were revealed after exposure to light for 4 h following 7 days in the dark. Among the genes that were differentially expressed by the presence of neuronal activity were several transcription factors (*Egr1*, *Pou3f1*, and *Erf*) that were specifically induced in OPCs. In addition, multiple OL populations displayed induction of the serum/glucocorticoid-regulated kinase, *Sgk1*, which was shown to be involved in oligodendrocyte process arborization [217]. Similarly, transcriptional changes in OPCs were also identified in the mouse cortex following seizure induction [218].

Therefore, neuronal activity appears to be a critical regulator of OPC proliferation, differentiation into OLs, and myelin formation through various messengers, receptors, and pathways. However, the extent to which neuronal activity directly influences epigenetic responses in OL remains understudied.

#### 7.2.2. Activity-Dependent Epigenetic Modifications in the CNS

Neurological disorders and behaviors are closely related to epigenetic changes. Activity-dependent regulation of the genome occurs in response to different environmental factors, such as learning, memory, and visual experience, and the epigenome is known to be altered in several neurological disorders, such as schizophrenia, Alzheimer’s disease, and multiple sclerosis [219,220,221,222,223]. While certain intracellular pathways and transcriptional changes have been correlated with neuronal activity, the mechanisms that link neuronal activation and these changes remain to be determined. Interestingly, it has been suggested that neuronal activity could modify the epigenetic landscape of neural cells, including oligodendroglial cells, as described in two studies. Further characterization of how these epigenetic changes may occur and impact oligodendroglial gene expression will need to be addressed by future experiments (Figure 3).

#### 7.2.3. Activity-Dependent Epigenetic Modifications in Oligodendrocytes

In 2014, Gibson et al. investigated how neuronal activity may regulate myelin-forming cells in vivo [22]. In this study, neuronal activity was induced by optogenetic stimulations in the premotor cortex of Thy1:channelrhodopsin 2 (ChR2) mice, in which ChR2-expressing neurons are activated by light [224] and elicited complex motor behavior in mice. The increased proliferation of OPCs was detected as early as 3 h after stimulation and resulted in a higher number of mature OLs and increased myelination 4 weeks after a 7-day stimulation paradigm. Notably, a concomitant increase in H3K9me3 and decrease in H3 acetylation was detected in dividing OPCs between 3 and 24 h after stimulation, suggesting that neuronal activity could enhance differentiation by inducing a repressive chromatin state that is necessary for OPC differentiation. Furthermore, administration of the histone deacetylase inhibitor TSA during the week of stimulation prevented mature oligodendrogenesis [22]. Chromatin in OLs can also be modified in response to social isolation, which has been associated with altered neuronal activity [225]. Indeed, in a mouse model of social isolation showing impaired myelination of the prefrontal cortex, immature chromatin and a lower level of heterochromatin formation was detected in the nucleus of an OL in the vicinity of axons with thinner myelin [177]. In agreement with Gibson’s study, the opposite changes in epigenetic marks were associated with chronic social isolation, namely, a decrease in repressive histone methylation and an increase in H3 acetylation. In parallel, changes in the level of transcripts for enzymes regulating these marks were detected, with significantly lower levels of *Hdac1* and *Hdac2* mRNAs [177].

Therefore, these two studies advocate for the role of neuronal activity regulation of the epigenome in OLs. One should consider that OPCs could differentially respond to neuronal activity, depending on their location, cell state, or age. For instance, region-specific changes in myelination have been detected in response to chronic stress [226]. While myelination deficits are observed in the nucleus accumbens of both susceptible and resilient mice, only the motor prefrontal cortex of susceptible mice show similar deficits, highlighting the distinct responses of oligodendroglial cells to chronic stress, depending on their spatial location. Moreover, recent single-cell studies have highlighted heterogeneity among the OL lineage in the healthy or diseased CNS, according to their location, but also their age. For example, the enrichment of oligodendroglial cells at mature stages, to the detriment of progenitor and immature stages, is detected as the brain ages [15,227,228,229]. This could be explained by different intrinsic properties in each oligodendroglial population, but also by the accumulation of distinct external and neuronal inputs throughout time. Of note, in zebrafish, functionally diverse subgroups of OPCs were identified in the spinal cord and were shown to differentially respond to neuronal activity [228].

#### 7.2.4. Activity-Dependent Epigenetic Modifications in Other Cell Types

Epigenetic modifications in response to neuronal activity have been identified in other cell types of the CNS, particularly in neurons. For instance, histone and DNA methylome modifications have both been detected in vivo following long-term potentiation in rats [230,231]. Neuronal activity can modulate the subcellular localization of histone deacetylases in neurons in vitro, following calcium influx through calcium channels and NMDA receptors, which could result in histone mark changes [232]. Indeed, membrane depolarization, which induced neural precursor cell differentiation, is associated with changes in histone methylation and acetylation marks [233]. In parallel, DNA demethylation in neurons in vitro has also been detected shortly after membrane depolarization and was associated with decreased H3K9me2 and increased histone acetylation in the regulatory region for brain-derived neurotrophic factor (BDNF), suggesting that DNA methylation changes may be associated with methylation-induced chromatin remodeling, which induces adaptive neuronal responses [234]. Furthermore, lower mRNA levels of DNMT1 and DNMT3a and higher levels of BDNF were detected in cortical neurons after depolarization by KCl and a sodium channel agonist [235]. DNA methylation changes have also been observed in epilepsy, which are characterized by abnormal patterns of neuronal activity, but also following electroconvulsive stimulation in mice [236,237]. Among DNMTs, only DNMT3a was higher after stimulation, making it a candidate for activity-dependent de novo methylation [237].

However, methylation changes observed in neurons might not reflect similar changes in glial cells. One study comparing hippocampal neuronal NeuN+ and glial NeuN− sorted cells following kainic acid injection, which elicits epilepsy, showed that there was almost no overlap in differentially methylated CpGs between neurons and non-neuronal cells [236]. Further investigation will be necessary to determine whether neurons and glial cells share common activity-dependent epigenetic responses.

## 8. Conclusions

As put forward by this review and others [83,238,239,240], the epigenome is critical in the regulation of the OL lineage, including DNA methylation, histone modifications, chromatin remodeling, lncRNA, and miRNAs. Further studies analyzing how distinct epigenetic layers might contribute together to produce gene regulation would be essential to more precisely characterize the role of epigenetics on the oligodendroglial cell lineage. Importantly, these mechanisms allow oligodendroglial cells to respond to their environment and, therefore, facilitate the appropriate response to maintain CNS function. Neuronal activity appears as an important modulator of OL biology, and we highlight here the few studies identifying activity-dependent epigenetic changes that can further regulate myelination. A deeper understanding of the link between neuronal activity and the epigenetic regulation of OLs would serve not only as a valuable resource to better understand neuroglia interaction but would also provide insights that could help in the development of therapies for neurodegenerative diseases, such as multiple sclerosis or Alzheimer’s disease, where the dysregulation of neuronal activity and myelin are often closely related.

## Figures and Tables

**Figure 1 life-11-00062-f001:**
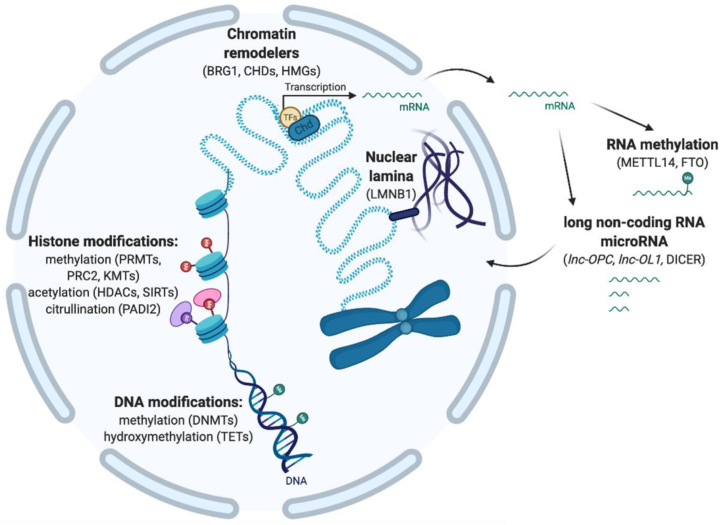
Several epigenetic layers are known to regulate gene expression, from DNA modification to post-translational histone modifications, chromatin remodeling, association with the nuclear lamina, and post-transcriptional regulation by RNA methylation and long non-coding and microRNA.

**Figure 2 life-11-00062-f002:**
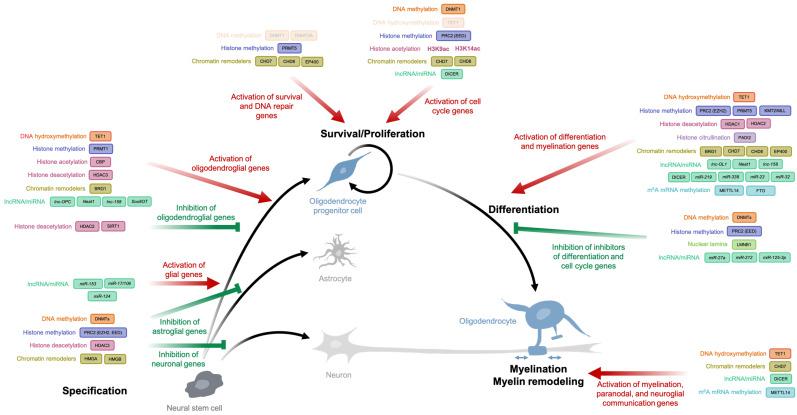
A combination of epigenetic modifications is involved in oligodendroglial cell specification, survival and proliferation, differentiation, and myelin remodeling. Red arrows represent activation and green arrows represent repression. Modifications and/or enzymes in faded font represent data that are still being discussed.

**Figure 3 life-11-00062-f003:**
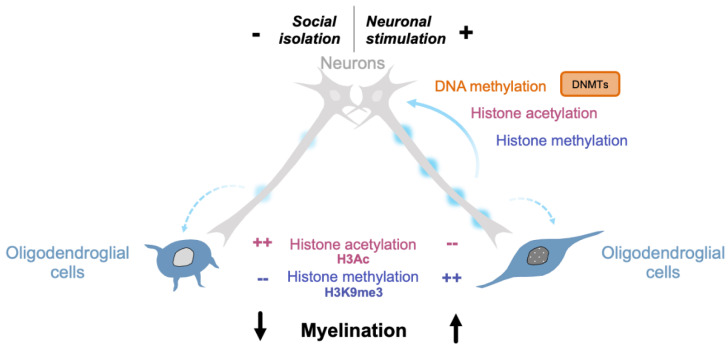
Neuronal activity modifies the epigenetic landscape of neural cells. Social isolation perturbs neuronal activity and reduces myelination. In this model, fewer heterochromatic nuclei are detected, along with increased histone acetylation and decreased histone methylation in OLs [177]; The stimulation of neuronal activity enhances myelination and is associated with a repressive chromatin state. OPC nuclei show decreased histone acetylation and increased histone methylation levels [22]; neuronal activity also modifies DNA methylation and histone acetylation and methylation marks in neurons.

**Table 2 life-11-00062-t002:** List of epigenetic marks and their roles during OPC survival and proliferation.

DNA Modification	Enzyme/Mark	Role	Targeted Genes or Functions	Model	Methods	References
DNA methylation	DNMT1/DNMT3A	Positive role in OPC survival and proliferation		In vitro OPCs + siRNA		Egawa et al. [132]
		No effect on OPC survival		Olig1cre;Dnmt1flox and Olig1cre;Dnmt3flox mice		Moyon et al. [91]
	DNMT1	Positive role in OPC proliferation	Increased DNA methylation and downregulation of cell cycle and cell proliferation genes (*Cdc6*, *Meis2*)	Olig1cre;Dnmt1flox mice	RNA-Seq and ERRBS on sorted neonatal OPCs and OLs, RNA-Seq on sorted Olig1cre;Tet1flox OPCs	Moyon et al. [91]
DNA demethylation	5cac mark		5cac enriched in glial cells during NSCs differentiation (*Gfap*, *Olig1/2*)	Descriptive study in vivo		Wheldon et al. [90]
DNA hydroxymethylation	TET1	Slight positive role during developmental OPC proliferation		Olig1cre;Tet1flox mice		Zhang et al. [92]
		No role in adult OPC proliferation		Olig1cre;Tet1flox and PdgfracreRT;Tet1flox mice		Moyon et al. [133]
** Histone Modification **	** Enzyme/Mark **	** Role **	** Targeted Genes or Functions **	** Model **	** Methods **	** References **
Histone methylation	PRC2 (EED) (H3K27me3)	Positive role in OPC proliferation (in brain, not spinal cord)		Olig1cre;Eedflox and PdgfracreRT;Eedflox mice		Wang et al. [96]
	PRMT5	Positive role in OPC survival	Inhibition of p53 pathway	Olig1cre;Prmt5flox mice and in vitro OPCs + siRNA	RNA-Seq on in vitro PRMT5-CRISPRKO OPCs	Scaglione et al. [134]
Histone (de)acetylation	H3K9ac/H3K14ac marks		Activation of cell cycle genes	In vitro OPCs + cMyc silencing		Magri et al. [135]
	HDACs	Positive role in OPC proliferation		In vitro O4^+^ cells + HDAC inhibitors		Conway et al. [136]
** Chromatin Organization **	** Enzyme/Mark **	** Role **	** Targeted Genes or Functions **	** Model **	** Methods **	** References **
Chromatin remodelers	CHD7	Positive role in OPC survival	Regulation of cell cycle genes (*Ccnd1*, *Cdk4* and *Cdk6*) and of cell survival/apoptosis genes (e.g., *p53/Trp53*, *Bax*, *Apaf1*)	PdgfracreRT;Chd7flox mice	RNA-Seq on in vitro PdgfracreRT;Chd7flox OPCs	Marie et al. [61]
		Positive role in OPC proliferation		PdgfracreRT;Chd7flox mice		Doi et al. [137]
		No role in OPC survival or proliferation		Olig1cre;Chd7flox mice		He et al. [59]
	CHD8	Positive role in adult OPC survival and proliferation in spinal cord		Olig1cre;Chd8flox and PdgfracreRT;Chd8flox mice		Zhao et al. [138]
	EP400	Positive role in OPC survival		Cnpcre;Ep400flox mice		Elsesser et al. [139]
** Post-Transcriptional Modification **	** Enzyme/Mark **	** Role **	** Targeted Genes or Functions **	** Model **	** Methods **	** References **
MicroRNA	DICER	Positive role in OPC proliferation		Olig1cre;Dicerflox mice		Zhao et al. [140]
m6A RNA methylation	PPRC2A	Positive role in OPC proliferation		Olig2cre;Pprc2aflox and Nescre;Pprc2aflox mice		Wu et al. [141]

ERRBS: enhanced reduced representation bisulfite sequencing, Seq: sequencing.

**Table 3 life-11-00062-t003:** List of epigenetic marks and their roles during OPC differentiation.

DNA Modification	Enzyme/Mark	Role	Targeted Genes or Functions	Model	Methods	References
DNA methylation	DNMT1	Positive role in OPC differentiation		In vitro OPCs + siRNA		Egawa et al. [132]
		Negative role in OPC differentiation and myelination	Decreased DNA methylation and upregulation of lipid synthesis and myelin formation genes (*Mog, Mag, Gpr37*)	Olig1cre;Dnmt1flox mice (no phenotype in Cnpcre;Dnmt1flox mice)	RNA-Seq and ERRBS on sorted neonatal OPCs and OLs, RNA-Seq on sorted Olig1cre;Dnmt1flox OPCs	Moyon et al. [91]
	DNMT1/DNMT3A	Positive role in OPC differentiation and remyelination		PlpcreRT;Dnmt1flox;Dnmt3flox mice		Moyon et al. [142]
DNA hydroxymethylation	TET1/TET2/TET3	Positive role in OPC differentiation		In vitro OPCs + siRNAs		Zhao et al. [35]
	TET1	Positive role in OPC differentiation and (re)myelination (no role for TET3)	Activation of cell differentiation genes (*Mag*)	Olig1cre;Tet1flox mice	hMeDIP-Seq on in vitro neonatal NSCs and OPCs, RNA-Seq on in vitro Olig1cre;Tet1flox OPCs	Zhang et al. [92]
		Positive role in OPC differentiation and adult remyelination only (no role for TET2)	Activation of cell differentiation, myelination, biosynthesis and neuroglial communication genes (*Slc* family members, *Pcdh* family members)	Olig1cre;Tet1flox and PdgfracreRT;Tet1flox mice	RNA-Seq and RRHP on sorted adult OPCs and OLs, RNA-Seq on oligo-enriched Olig1cre;Tet1flox lesioned spinal cord	Moyon et al. [133]
** Histone Modification **	** Enzyme/Mark **	** Role **	** Targeted Genes or Functions **	** Model **	** Methods **	** References **
Histone methylation	PRC2 (EED) (H3K27me3)	Negative role in OPC differentiation and (re)myelination	Chromatin silencing of differentiation genes (*Bmp, Wnt*)	Olig1cre;Eedflox, Plpcre-Eedflox and PdgfracreRT;Eedflox mice	RNA-Seq and ChIP-Seq on in vitro Olig1cre;Eedflox OPCs	Wang et al. [96]
	H3K9me3	Positive role in OPC differentiation (no role for H3K27me3)		In vitro OPCs and OLs	ChIP-Seq on in vitro OPCs and Ols	Liu et al. [97]
	KMT2/MLL (H3K4me3)	Positive role in OPC differentiation	Recruited by CHD8, mark accumulation at promoters of differentiation genes (*Tcf7l2, Myrf, Zfp488, Lnc-OL1*)	Olig1cre;Chd8flox mice + KDM5 demethylase inhibitors	ChIP-Seq on in vitro Olig1cre;Chd8flox OPCs	Zhao et al. [138]
	PRMT5	Positive role in OPC differentiation and myelination	Regulation of OPC differentiation genes	Olig1cre;Prmt5flox mice and in vitro OPCs + siRNA	RNA-Seq on in vitro PRMT5-CRISPRKO OPCs	Scaglione et al. [134]
		Positive role in OPC differentiation and myelination	Regulation of OPC differentiation genes	Olig2cre;Prmt5flox mice	RNA-Seq on Olig2cre;Prmt5flox brain	Calabretta et al. [154]
		Positive role in OPC differentiation	Repression of inhibitors of differentiation (*Id2/4*)	In vitro OPCs and glioma cells + siRNA		Huang et al. [155]
Histone deacetylation	HDACs	Positive role in OPC differentiation		In vitro O4^+^ cells + HDAC inhibitors		Conway et al. [136]
				In vitro OPCs + HDAC inhibitors		Marin-Husstege et al. [143]
		Positive role in OPC differentiation and myelination		In vivo HDAC inhibitors		Shen et al. [156]
		Positive role in OPC differentiation and remyelination				Shen et al. [157]
	HDAC1/HDAC2	Positive role in OPC differentiation and myelination		Olig1cre;Hdac1flox;Hdac2flox mice		Ye et al. [121]
	HDAC1	Positive role in OPC differentiation		In vitro OPCs + siRNA		Egawa et al. [132]
	HDAC2	Negative role in OPC differentiation				
	HDAC6	Positive role in OPC differentiation and OL morphology	Deacetylation of alpha-tubulin	In vitro OPCs + siRNA		Noack et al. [158]
	HDAC11	Positive role in OPC differentiation		In vitro Olineu cells + siRNA		Liu et al. [159]
	SIRT2	Negative role in OPC differentiation	Deacetylation of alpha-tubulin	In vitro Olineu cells + siRNA		Li et al. [160]
Histone citrullination	PADI2	Positive role in OPC differentiation	Activation of cell differentiation genes (*Septins, Mbp, Sox9/10, Tcf7l2*) (+ role on non-histone targets, e.g., myelin proteins)	PdgfracreRT;Padi2flox mice, in vitro OPCs + siRNA, in vitro Olineu + overexpression	Proteomic and ATAC-Seq on in vitro OPCs + siRNA	Falcão et al. [55]
** Chromatin Organization **	** Enzyme/Mark **	** Role **	** Targeted Genes or Functions **	** Model **	** Methods **	** References **
Chromatin remodelers	CHD7	Positive role in OPC differentiation and (re)myelination	Transcriptional activation of differentiation genes (*Sox10, Gpr17, Sirt2, Nkx2.2*)	PdgfracreRT;Chd7flox mice	RNA-Seq on in vitro PdgfracreRT;Chd7flox OPCs	Marie et al. [61]
				PdgfracreRT;Chd7flox mice		Doi et al. [137]
			Transcriptional activation of differentiation genes (*Myrf, Sox10*)	Olig1cre;Chd7flox mice	RNA-Seq on Olig1cre;Chd7flox spinal cord	He et al. [59]
	CHD8	Positive role in OPC differentiation and (re)myelination	Transcriptional activation of differentiation genes (*Tcf7l2, Myrf, Zfp488, Lnc-OL1*)	Olig1cre;Chd8flox and PdgfracreRT;Chd8flox mice	ChIP-Seq and ATAC-Seq on in vitro Olig1cre;Chd8flox OPCs	Zhao et al. [138]
	BRG1	Positive role in OPC differentiation and myelination	Transcriptional activation of differentiation genes (*Myrf, Sox10*)	Olig1cre;Brg1flox mice	RNA-Seq and ChIP-Seq on Olig1cre;Brg1flox optic nerve	Yu et al. [62]
				Cnpcre;Brg1flox mice		Bischof et al. [57]
	EP400	Positive role in OPC differentiation	Transcriptional activation of differentiation genes (*Plp1, Mbp, Mog, Myrf, Sox10*)	Cnpcre;Ep400flox mice	ChIP-Seq on sorted Cnpcre;Ep400flox OPCs	Elsesser et al. [139]
Nuclear lamina	LMNB1	Negative role in OPC differentiation	Repression of differentiation genes (cholesterol synthesis, *Lss*)	In vitro OPCs + LMNB1 overexpression	DamID on in vitro OPCs, combining with LMNB1 maintained expression	Yattah et al. [68]
** Post-Transcriptional Modification **	** Enzyme/Mark **	** Role **	** Targeted Genes or Functions **	** Model **	** Methods **	** References **
Long non-coding RNA	*lnc-OL1*	Positive role in OPC differentiation and (re)myelination	Silencing of OPC program during their differentiation (interaction with SUZ12, within PRC2)	In vitro OPCs + lnc-OL1 overexpression and Ezh2 siRNA, Olig1cre;Ezh2flox mice		He et al. [161]
	*Neat1*	Positive role in OPC differentiation	Over-representation of oligodendroglial pathways among differentially expressed genes	Neat1−/− mice	RNA-Seq of Neat1−/− mouse brains	Katsel et al. [115]
	*lnc-158*	Positive role in OPC differentiation		In vitro NSCs + lnc-158 overexpression and siRNA		Li et al. [113]
MicroRNA	DICER	Positive role in OPC differentiation and myelination		Olig2cre;Dicerflox and Cnpcre;Dicerflox mice		Dugas et al. [162]
		Positive role in myelination		Olig1cre;Dicerflox mice		Zhao et al. [140]
	*miR-219*	Positive role in OPC differentiation	Repression of inhibitors of differentiation (*Zfp238*, *Foxj3*)	In vitro OPCs + miR-219 overexpression		Dugas et al. [162]
	*miR-338*	Positive role in OPC differentiation	Repression of inhibitors of differentiation (*Sox6*, *Hes5*)	In vitro OPCs + miR-219 overexpression, in vivo electroporation in chicks and zebrafish		Zhao et al. [140]
	*miR-23*	Positive role in OPC differentiation and myelination	Upregulation of myelin genes (*Cnp*, *Plp1*, *Mag*, *Mog*) and downregulation of OPC and nuclear lamina genes (*Pdgfra*, *Lmnb1*)	In vivo miR-23 overexpression in CNP^+^ cells (mouse model)	RNA-Seq	Lin et al. [163]
	*miR-32*	Positive role in OPC differentiation	Promotes expression of myelin proteins and glucose/lipi metabolism (MBP, SLC45A3)	In vitro OPCs + miR-32 overexpression and shRNA		Shin et al. [164]
	*miR-27a*	Negative role in OPC differentiation and (re)myelination	Targets mature OL-specific genes (*Mbp*)	In vitro OPCs + miR-27a overexpression, ex vivo inhibition in cerebellum slices, in vivo intranasal inhibition in mice	RNA-Seq	Tripathi et al. [165]
	*miR-212*	Negative role in OPC differentiation	Inhibition of oligodendroglial genes (*Mbp*, *Olig1/2*, *Sox10*)	In vitro OPCs + miR-212 overexpression and siRNA	Validation by rtqPCR	Wang et al. [166]
	*miR-125-3p*	Negative role in OPC differentiation	Inhibition of differentiation genes (*Fyn*, *Smad4*, *Nrg1*)	In vitro OPCs + miR-125-3p overexpression and siRNA	Validation by rtqPCR	Lecca et al. [167]
m6A RNA methylation	METTL14	Positive role in OPC differentiation	Necessary for mature myelin gene expression (*Mbp, Mog, Mag, Plp1, Cnp*) and regulation of histone modification enzymes (HATs, HMTs, HDACs, KDMs)	Olig2cre;Mettl14flox and Cnpcre;Mettl14flox mice	RNA-Seq on in vitro Olig2cre;Mettl14flox OPCs	Xu et al. [168]
	FTO	Positive role in OPC differentiation and myelination	Promotes *Olig2* degradation	Fto-TG mice	RNA-Seq on Nescre;Prrc2aflox brain and RIP-Seq on Olig2cre;Prrc2aflox brain	Wu et al. [141]

ATAC: assay for transposase-accessible chromatin, ChIP: chromatin immunoprecipitation, DamID: DNA adenine methyltransferase identification, ERRBS: enhanced reduced representation bisulfite sequencing, hMeDIP: hydroxyMethylated DNA immunoPrecipitation, RIP: RNA immunoprecipitation, RRHP: reduced representation 5-hydroxymethylcytosine profiling, rtqPCR: reverse transcription quantitative polymerase chain reaction, Seq: sequencing.

**Table 4 life-11-00062-t004:** List of epigenetic marks and their roles during OPC myelination.

DNA Modification	Enzyme/Mark	Role	Targeted Genes or Functions	Model	Methods	References
DNA hydroxymethylation	TET1	Positive role in myelination and neuroglial communication	Activation of cell differentiation and cell communication genes (Ca^2+^ homeostasis)	Olig1cre;Tet1flox mice	hMeDIP-Seq on in vitro neonatal NSCs and OPCs, RNA-Seq on in vitro Olig1cre;Tet1flox OPCs	Zhang et al. [92]
			Activation of cell differentiation, myelination, biosynthesis, and neuroglial communication genes (*Slc* family members, *Pcdh* family members)	Olig1cre;Tet1flox mice	RNA-Seq and RRHP on sorted adult OPCs and OLs, RNA-Seq on oligo-enriched Olig1cre;Tet1flox lesioned spinal cord	Moyon et al. [133]
** Chromatin Organization **	** Enzyme/Mark **	** Role **	** Targeted Genes or Functions **	** Model **	** Methods **	** References **
Chromatin remodelers	CHD7	Positive role in myelination	Preferentially binding and increased accessibility of myelinogenesis (*Mbp, Plp1* and *Cnp*) and lipid metabolism genes (*Enpp2*, *Nfya,* and *Elovl7*)	Olig1cre;Chd7flox mice	RNA-Seq on in vitro Olig1cre;Chd7flox spinal cord	He et al. [59]
** Post-Transcriptional Modification **	** Enzyme/Mark **	** Role **	** Targeted Genes or Functions **	** Model **	** Methods **	** References **
MicroRNA	DICER	Positive role in myelin maintenance	Induces expression of myelin proteins	PlpcreRT;Dicerflox mice		Shin et al. [173]
m6A RNA methylation	METTL14	Positive role in myelination and regulation of internodes lenghts	Alternative splicing of glial paranodal genes (*Nfasc155*)	Olig2cre;Mettl14flox mice	RNA-Seq on in vitro Olig2cre;Mettl14flox OPCs and OLs	Xu et al. [168]

hMeDIP: hydroxymethylated DNA immunoprecipitation, RRHP: reduced representation 5-hydroxymethylcytosine profiling, Seq: sequencing.

## Data Availability

Data sharing not applicable.

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
