# Peer review of "Oligodendroglial Epigenetics, from Lineage Specification to Activity-Dependent Myelination"

_life, 2021, doi:10.3390/life11010062_

Round 1

Reviewer 1 Report

This review well describes current knowledge regarding the epigenome and its critical role in the regulation of  differentiation of NPS up to  oligodendrocytes. It details the main proteins involved in DNA methylation, histone modification, chromatin remodelling as well as the function of miRNA and lncRNA. I find it interesting and informative. Since the field is quite complex and dispersed it would be very useful for the reader who is not familiar with these aspects of oligidendrogenesis if one figure could be dedicated to highlight the main epigenetic events that occur, to show the most important modifications that mark the passage from one stage of differentiation to the next.

Minor points:

The text in the figures and tables is too small 

Author Response

We would like to thank the reviewer for their overall comments on our review.

At the moment, the literature does not allow us to precise which epigenetic modification might be more or less important for oligodendroglial cell lineage. We now mention this lack of knowledge in the field in our conclusion (l.580-582): “Further studies analyzing how distinct epigenetic layers might contribute together to gene regulation would be essential to more precisely characterize the role of epigenetics on the oligodendroglial cell lineage”. We have tried to reorganize the figure 2, to highlight the role of each epigenetic mark at the each oligodendroglial stage: specification, survival, proliferation, differentiation and myelination. We have also added which pathways are regulated by these marks at each stage.

As requested, we have increased the font size in each table and figure.

You will find attached our revised manuscript. Changes are underlined in this new version.

Reviewer 2 Report

In general, this is a well-written review summarizing multiple findings linked to the epigenetics of oligodendroglial differentiation and myelination processes. Nevertheless, I recommend some improvements before publication, as listed below.

  1. Line 23 - Ref Valerio-Gomes et al. 2018 seems to refer to mice brain, not human. Could you please make an appropriate correction or comment?
  2. Line 92, 186, Figure 2 – as far as I understand, SUZ12 and EED, which are indeed part of the PRC2 complex responsible for H3K27 trimethylation, do not possess lysine methyltransferase activity. Please be more accurate. There are many more HMTs (MLLs, SETDB1, SUV39H1/2, etc) that methylate other lysines within the H3 chain, which could be mentioned.
  3. Line 104 – please note that enhancer regions are usually identified by H3K27ac and H3K4me1 marks. Please be more specific in your manuscript.
  4. Line 108 – please note that histone citrullination reduces the proteins' charge, thus lowering the interaction between DNA and histones. As such, histone citrullination also leads to chromatin decondensation and gene overexpression. Please be more specific in your manuscript.
  5. Figure 2 – in some instances, the font is very difficult to read. I suggest increasing its size and maybe changing the font color to black. It would also be useful if the same epigenetic modifications were color-coded in the same way across the figure (e.g., DNA hydroxymethylation/methylation have different colors). Please provide a name for the cell type on the right side.
  6. Table 1 – increased font size will be helpful. Is it possible to add some references to Table 1? The first column with epigenetic modification could be narrower, while more space could be provided to other columns (e.g., three last columns). Please specify the role of the specific epigenetic modifier (e.g., promotes or suppresses differentiation?).
  7. Line 243 – it seems there is some mistake with word order. Please re-phrase for better clarity.
  8. Lines 244-247 – could you please provide more details on the roles of listed lncRNAs?
  9. Table 2, 3,4 – please increase the font size. Please specify the roles of epigenetic modifiers (e.g., increases or decreases survival?). Please add references to Table 3.
  10. Line 384 – could you provide some examples of relevant OPC–specific inhibitors of differentiation and cell cycle genes?
  11. Point 7.2.3. the second paragraph is a bit vague. Some specific examples of "region-specific changes in myelination", "heterogeneity among OL lineage" would be beneficial, if available.

Author Response

We would like to thank the reviewer for their overall comments on our review.

1/ Indeed, our reference [1] was referring to mouse brain data. We have now corrected this sentence and added human data quantification to our statement (l.23-25): “Oligodendrocytes (OLs) are glial cells of the central nervous system (CNS) representing 20% of cells in adult mouse brain and even up to 40% of neural cells in human neocortical regions.”

2/ As noticed by the reviewer, SUZ12 and EED are non-catalytic parts of the PRC2 complex, which has a methyltransferase activity. We have clarified their descriptions (l.95-97): “SUZ12, EZH2 and EED, which are parts of the polycomb repressive complex 2, PRC2, in oligodendroglial cells”. We also now mention PRC2 (SUZ12) and PRC2 (EED) in tables and figures, when we refer to the functional characterization of PRC2, using Suz12 or Eed loss-of-function, respectively.

For sack of clarity, we made the choice to mainly focus on enzymes or enzyme components that have been functionally studied in the oligodendroglial lineage. We now briefly introduce SETDB1 and SUV39H1/2 in the first part, as they have been shown to be differentially expressed in the oligodendroglial lineage (l.96-97): “Histone methylation of lysine residues is catalyzed by histone lysine methyltransferases (HMTs) (e.g. KMT2/MLL, SETDB1, SV39H1/2”. We were previously mentioning the lysine methyltransferase KMT2/MLL, but without naming it. We have now introduced it in the first part, and identified its role in OPC differentiation (l.354-356): “CHD8 can itself directly recruit KMT2/MLL, an histone post translational modifying enzymes responsible for the addition of H3K4me3”, as well as in Figure 2 and Table 3.

3/ We have modified our manuscript and precise the identification of enhancers (l.104-107): “H3K27ac and H3K4me1, as well as H3K4me3 have been strongly associated with gene activation at enhancers and transcription start sites, respectively”.

4/ Indeed, we have now detailed the dual role of citrullination (l. 107-111): “Histone arginine citrullination, mainly catalyzed by peptidylarginine deiminase 2 (PADI2) in oligodendroglial cells, has a dual role of gene regulation. Citrullination is directly associated with gene activation, as it reduces proteins charge and histone-DNA histone interaction, favoring chromatin decondensation, while indirectly, it also blocks repressive methylation on arginine residues”.

5/ Following the reviewer’s advice, we have increased the font size, change the font color of the enzymes in black, change hydroxymethylation color to the same as methylation. The cell on the right side was named “Oligodendrocyte”, but we have also increased the font to be more visible. We have also extended and enlarged the overall figure to have a clearer image.

6/ & 9/ We have also increased the font size for all tables. The references were already included in each table, but we have now updated our manuscript following Life format and verify that all full tables are displayed. For each mark or enzyme, we have detailed if they have a positive or negative role for oligodendroglial specification, proliferation, differentiation, myelination.

7/ We have tried to clarify this sentence (l. 240-241): “They identified lnc-OPC, a specific and highly lncRNA expressed in OPCs, which is critical for cell fate determination”.

8/ We have now detailed the role of the following lncRNAs on OPC specification (l.241-244): “In vitro loss- and gain-of-functions experiments, targeting lnc-OPC, and also Sox8OT, Neat1 and lnc-158, have highlighted the positive prominent role of lncRNAs for oligodendroglial specification.”

10/ We have added examples of genes negatively regulated during OPC differentiation (l. 362-365): “These marks mainly regulate the downregulation of OPC-specific inhibitors of differentiation (such as Id2/Id4) or cell cycle (such as Cdk4/6, Cxcl2/5/10/14) genes, and therefore are often essential for OPC cell cycle exit and early OL differentiation, but less involved in myelin maintenance”.

11/ We have also detailed this paragraph and added a few examples (l. 534-546): “These two studies therefore advocate for a role of neuronal activity regulation of the epigenome in OLs. One should consider that OPCs could differentially respond to neuronal activity depending on their location, cell state or age. For instance, region-specific changes in myelination have been detected in response to chronic stress. While myelination deficits are observed in the nucleus accumbens of both susceptible and resilient mice, only the motor prefrontal cortex of susceptible mice show similar deficits, highlighting distinct responses of oligodendroglial cells to chronic stress depending on their spatial location. Moreover, recent single-cell studies have highlighted heterogeneity among the OL lineage in the healthy or diseased CNS, according to their location, but also to age. For example, an enrichment of oligodendroglial cells at mature stages, to the detriment of progenitor and immature stages, is detected as the brain ages. This could be explained by different intrinsic properties in each oligodendroglial population, but also by the accumulation of distinct external and neuronal inputs throughout time”.

You will find attached our revised manuscript. Changes are underlined in this new version.
